# The Role of Calcium-Independent Phospholipase A_2_ in the Molecular Mechanisms of Schizophrenia

**DOI:** 10.3390/cells14171348

**Published:** 2025-08-30

**Authors:** Shoji Nakamura

**Affiliations:** Graduate School of Medicine, Yamaguchi University, Yamaguchi 755-8505, Japan; snaka@yic.ac.jp

**Keywords:** schizophrenia, psychosis, phospholipase A_2_, neurodegenerative disease, neuroinflammation

## Abstract

Schizophrenia, depression, and bipolar disorder may represent neurodegenerative conditions involving both degeneration and aberrant regeneration of monoaminergic axons. Negative and cognitive symptoms could arise from monoaminergic axon degeneration, whereas positive symptoms and manic states might result from excessive axonal regeneration and sprouting. The molecular mechanisms driving these opposing processes remain largely unclear. This review considers the possible role of calcium-independent phospholipase A_2_ (iPLA2) in regulating monoamine axon degeneration and hyper-regeneration in schizophrenia. Emerging evidence suggests that pro-inflammatory signaling mediated by cytosolic PLA_2_ (cPLA2) may promote monoamine axon degeneration, while anti-inflammatory iPLA2 activity could facilitate regeneration and sprouting. Overactivation of iPLA2 might lead to aberrant axonal sprouting, potentially contributing to positive symptoms through hyperdopaminergic states in the medial prefrontal cortex (mPFC). Conversely, axon degeneration in the same region may underlie negative and cognitive symptoms. The review also discusses a potential interplay between dopamine and N-methyl-D-aspartate (NMDA) receptor signaling in distinct neuronal populations of the mPFC and suggests that targeting iPLA2 and its pathways could represent a promising therapeutic strategy. Viewing schizophrenia and related disorders through the lens of monoamine axon pathology may eventually improve diagnostic precision and inform the development of treatments aimed at restoring the balance between degeneration and regeneration.

## 1. Introduction

The major mental disorders—schizophrenia, major depression, and bipolar disorder—are closely linked to dysfunctions in monoamine neurotransmitters: dopamine (DA), noradrenaline (NA), and serotonin (5-HT). Hypo-monoaminergic states are associated with depressive and negative/cognitive symptoms, whereas hyper-monoaminergic states underlie positive and manic symptoms. However, the mechanisms driving these opposing states remain unclear. Notably, monoamine axons in the adult brain exhibit a remarkable ability to regenerate spontaneously after degeneration and, in some cases, can undergo excessive regeneration or sprouting [1,2,3,4,5,6]. Based on this capacity, it is conceivable that hypo-monoaminergic states may result from monoamine axon degeneration, whereas hyper-monoaminergic states could stem from excessive regeneration and aberrant sprouting of these axons [7]. Accordingly, schizophrenia and bipolar disorder can be hypothetically conceptualized as monoamine axon disorders involving both degeneration and excessive regeneration of axons, in the absence of neuronal cell loss. Given the possible link between positive symptoms and hyperdopaminergic states, schizophrenia is thought to involve excessive regeneration of DA axons following their degeneration.

The processes governing monoamine axon degeneration and regeneration are thought to be influenced by pro- and anti-inflammatory mechanisms. Specifically, pro-inflammatory stressors drive monoamine axon degeneration, contributing to the pathophysiology of schizophrenia, depression, and bipolar disorder, while anti-inflammatory responses—including axonal regeneration—are associated with recovery from these conditions [7,8]. Emerging evidence has further implicated the phospholipase A_2_ (PLA2) signaling pathway, which plays a pivotal role in both pro- and anti-inflammatory responses [9,10], in the regulation of axonal degeneration and regeneration in both peripheral and central nervous system neurons, including monoamine axons [11,12,13,14].

In this review, I examine the role of the PLA2-signaling pathway in the molecular mechanisms driving hyper-regeneration and sprouting of monoaminergic axons, processes that may underlie the positive symptoms of schizophrenia. I also discuss the brain regions that may play a central role in the emergence of these symptoms. Finally, I offer novel insights into the diagnosis and treatment of monoaminergic axon disorders, with particular emphasis on the contribution of the PLA2 pathway—especially calcium-independent PLA2 (iPLA2)—to the pathophysiology of schizophrenia.

## 2. Cortical Atrophy and Negative/Cognitive Symptoms in Schizophrenia

Patients with schizophrenia exhibit cortical gray matter shrinkage and ventricular enlargement due to cortical degeneration, which is primarily associated with negative and cognitive symptoms rather than positive symptoms [15,16]. Evidence suggests that this cortical atrophy results from reduced synaptic connectivity rather than neuronal cell loss [17,18]. The degeneration of cortical DA axons aligns with cortical atrophy in schizophrenia and is thought to contribute to the manifestation of negative and cognitive symptoms.

A postmortem study provided evidence of DA axon degeneration in the medial prefrontal cortex (mPFC) of individuals with schizophrenia [19]. In this study, the density of DA axons in the dorsomedial PFC was analyzed in 16 pairs of schizophrenia patients and matched controls. Compared to controls, ten schizophrenia patients exhibited a clear reduction in DA axon density, while two showed an increase, and four remained unchanged. In contrast, no significant difference was observed in the density of cortical serotonergic axons between schizophrenia patients and controls. These findings suggest the involvement of DA axon degeneration in the mPFC in a subset of schizophrenia patients, potentially contributing to their negative and cognitive symptoms.

Furthermore, given that excessive regeneration and sprouting of monoamine axons can follow initial degeneration, it is plausible that DA axon degeneration in the mPFC sometimes may trigger hyper-regeneration and sprouting, contributing to the development of positive symptoms. This suggests that DA axons in the mPFC may play a role in the pathophysiology of both negative/cognitive and positive symptoms in schizophrenia.

## 3. Hyper-Regeneration/Sprouting and Positive Symptoms: Involvement of Calcium-Independent PLA2

Various stressors, including psychosocial stress, can induce neuronal damage through neuroinflammation, leading to the release of pro-inflammatory cytokines (e.g., IL-1β, IL-6, and TNF-α) and mediators such as arachidonic acid (AA) metabolites [20,21,22,23,24]. Persistent pro-inflammatory stress can drive neurodegenerative changes, including axonal degeneration. Cytosolic PLA2 (cPLA2) is known to release AA from the cell membrane, where its metabolites play a central role in pro-inflammatory signaling. In contrast, iPLA2 is involved in anti-inflammatory responses by releasing anti-inflammatory mediators such as eicosapentaenoic acid (EPA) and docosahexaenoic acid (DHA) from membrane phospholipids [9,10]. The iPLA2-signaling pathway has been reported to play a critical role in axonal regeneration and sprouting in both peripheral and central neurons, including monoaminergic axons [12,13,14,25,26,27].

Given this, it is plausible that overactivation of the iPLA2-signaling pathway contributes to the hyper-regeneration and sprouting of monoamine axons, potentially leading to the positive symptoms of schizophrenia and bipolar mania. Conversely, activation of pro-inflammatory cPLA2 may drive monoamine axon degeneration, thereby contributing to the negative and cognitive symptoms observed in schizophrenia, depression, and bipolar depression (Figure 1).

Interestingly, elevated iPLA2 activity has been reported in the serum of schizophrenia patients, and higher serum iPLA2 activity has been linked to an attenuated niacin skin flushing response in these individuals [28,29]. Notably, a similar response has been observed in nonpsychotic first-degree relatives of schizophrenia patients [30], further suggesting a genetic association between iPLA2 and schizophrenia. Additionally, paranoid schizophrenia patients have been found to exhibit excessive terminal branching of motor neuron axons innervating skeletal muscles more frequently than nonparanoid schizophrenia patients and healthy controls [31,32].

Given the role of iPLA2 in axonal regeneration and sprouting in both the peripheral and central nervous systems, it is plausible that iPLA2 overactivation in schizophrenia—particularly in patients exhibiting positive symptoms—leads to excessive axonal branching across multiple neural circuits. Aberrant axonal sprouting may not be limited to monoaminergic neurons but could also involve non-monoaminergic projections, such as glutamatergic and GABAergic axons. Such changes may contribute to the heterogeneous nature of positive symptoms in psychiatric disorders. The potential involvement of hyperbranching in non-monoaminergic axons could further complicate the disorder’s symptomatology. However, the extent to which these aberrant non-monoaminergic projections may contribute to the development of positive symptoms remains unclear. Moreover, iPLA2-mediated hyperinnervation of serotonergic and noradrenergic axons could underlie the frequent co-occurrence of manic features observed in some patients with schizophrenia. Since axon hyperbranching is thought to follow axonal degeneration, degeneration of non-monoaminergic axons may also contribute to the cortical atrophy, via synaptic loss, commonly observed in schizophrenia.

## 4. Hyper-Regeneration/Sprouting of Monoamine Axons in Peripheral Tissues

Hyper-regeneration and sprouting of monoamine axons have been observed not only in the brain but also in peripheral organs. For instance, following myocardial infarction, excessive sympathetic NA axon innervation occurs in the infarcted heart, contributing to arrhythmias [33,34]. Similar sympathetic hyperinnervation has also been reported in other tissues, including smooth muscle and lymphoid tissues [35,36,37]. In addition, parasympathetic cholinergic axons, as well as sympathetic axons, have been shown to exhibit hyperinnervation in cardiac tissues following continuous electrical stimulation of the stellate ganglion in normal dogs [38]. This suggests that electrical activation of neurons induces the release of neurotrophic factors, which in turn promote axonal sprouting and branching.

Nerve growth factor (NGF) has been implicated in driving sympathetic hyperinnervation [33,34,35], while the glycogen synthase kinase-3 (GSK3) inhibitor lithium (Li) has been shown to prevent this process by attenuating NGF production [34]. Notably, Li exhibits dual effects on axonal growth and regeneration, promoting these processes at low concentrations while inhibiting them at higher concentrations [39,40]. This duality is thought to underlie its therapeutic effects in bipolar disorder, facilitating axonal growth during depressive episodes and inhibiting excessive sprouting during manic episodes [7].

Further supporting the role of lipid signaling in axonal regeneration, studies in animal models of corneal nerve damage have demonstrated that iPLA2 activation facilitates nerve regeneration by releasing DHA from membrane phospholipids, leading to increased synthesis of docosanoids, such as neuroprotectin D1 (NPD1) and resolvin D6 [14]. Importantly, NPD1 also enhances the synthesis of multiple neurotrophic factors, including NGF and brain-derived neurotrophic factor, highlighting the role of the iPLA2 signaling pathway in neurotrophic support across both peripheral and central nervous system axons.

Based on these findings, the inhibitory effects of Li (via GSK3 inactivation) on sympathetic hyperinnervation are likely mediated by the suppression of iPLA2-induced NGF production. This suggests a broader role for lipid-signaling pathways in axonal remodeling and their potential relevance to neuropsychiatric disorders. However, the extent to which these peripheral phenomena contribute to the pathophysiology of schizophrenia remains to be determined.

## 5. The Involvement of mPFC Neurons in the Occurrence of Positive and Negative/Cognitive Symptoms via Inactivation of DA-Mediated NMDA Responses

As previously discussed, the mPFC may be the key brain region underlying the positive symptoms of schizophrenia, rather than the striatum, which was traditionally considered the primary site [7]. Notably, neonatal hypoxia has been shown to induce the neurotrophic factor neuregulin-1 (NRG1) [41], and a mouse model of schizophrenia—generated by neonatal administration of NRG1—exhibited hyperinnervation of DA axons in the mPFC but not in the striatum [42]. These mice also displayed behavioral abnormalities reminiscent of schizophrenia’s positive symptoms, including impaired prepulse inhibition and latent inhibition, reduced social interaction, and heightened sensitivity to amphetamine. This model provides compelling evidence that DA axon hyperinnervation in the mPFC contributes to behaviors analogous to the positive symptoms of schizophrenia. Furthermore, it demonstrates that schizophrenia-like behaviors can arise following neonatal insults, supporting the neurodevelopmental hypothesis that disruptions in brain development—potentially occurring perinatally—contribute to the onset of schizophrenia symptoms later in life. However, whether iPLA2 contributes to DA axon hyperinnervation in this model remains to be elucidated, although iPLA2 knockout has been reported to cause degeneration of DA axons [13].

Further studies suggest that mPFC neurons are involved in both positive and negative/cognitive symptoms through mechanisms involving N-methyl-D-aspartate (NMDA) and DA receptor interactions. Pyramidal neurons in the mPFC are divided into two distinct populations: one expressing the dopamine D1 receptor (D1R) and the other expressing the dopamine D2 receptor (D2R) [43,44]. Importantly, DA modulates NMDA receptor function in these neurons in a concentration-dependent manner: low DA levels enhance NMDA activity via D1R, whereas high DA levels inhibit NMDA activity via D2R [45]. This suggests that D1R-mediated NMDA activation and D2R-mediated NMDA inhibition occur independently within distinct neuronal populations.

The degeneration of DA axons in the mPFC may contribute to negative/cognitive symptoms by impairing both D1R- and D2R-mediated NMDA functions due to DA deficiency. Conversely, DA axon hyper-regeneration and excessive sprouting could lead to positive symptoms through D2R-mediated hyper-inhibition of NMDA activity, driven by excessive DA release (Figure 2). This model of NMDA hypofunction aligns with the effects of NMDA receptor antagonists such as phencyclidine and ketamine, which can induce schizophrenia-like positive symptoms along with negative/cognitive symptoms via distinct neuronal pathways [7,46,47].

## 6. Diagnosis and Treatment of Schizophrenia, Depression, and Bipolar Disorder as Monoamine Axon Disorders

Schizophrenia, major depression, and bipolar disorder are hypothesized to be neurodegenerative conditions associated with the degeneration of monoaminergic axons. This degeneration, often followed by aberrant hyper-regeneration or sprouting, may be driven by persistent pro-inflammatory stress. To directly assess which monoamine axons are affected in an individual patient, positron emission tomography (PET) imaging with radiolabeled monoamine transporter ligands is essential. This approach enables the quantification of monoamine axon density, allowing for precise identification of the affected monoamine system(s) and brain region(s), thereby facilitating targeted therapeutic interventions.

In monoaminergic axon disorders, as in Parkinson’s disease, degeneration typically begins at the distal axon terminals and progresses toward the cell body [48,49,50]. Distal axons are particularly vulnerable to damage compared to dendrites and soma, as they are the farthest from the cell body, which supplies critical anti-inflammatory factors necessary to prevent degeneration. Consequently, the earliest symptoms of neurodegenerative diseases are more likely to arise from distal axon degeneration rather than from impairments in dendrites or soma. Moreover, it is noted that axonal length and the extent of terminal arborization may play a pivotal role in neuronal vulnerability. Neurons with longer axons and more extensive branching are more prone to cell death [8].

Given the remarkable capacity of monoaminergic axons for spontaneous regeneration and sprouting, the severity and progression of these disorders may depend on the balance between pro- and anti-inflammatory responses or the interplay between degenerative and regenerative mechanisms (Figure 3). If anti-inflammatory responses outweigh pro-inflammatory processes, monoamine axon degeneration may be suppressed, allowing for damaged axons to initiate regenerative processes. Thus, reducing pro-inflammatory stress represents a straightforward and potentially effective strategy for promoting recovery from monoamine axon disorders such as schizophrenia.

As discussed previously, psychosocial stress is a major contributor to monoamine axon degeneration, as it activates pro-inflammatory signaling pathways and induces the release of pro-inflammatory mediators. Identifying and mitigating such stressors is therefore crucial for preventing neuroinflammation and the subsequent degeneration of monoamine axons. While NMDA receptor agonists may be effective in alleviating both negative/cognitive and positive symptoms of schizophrenia, they do not address the underlying pathological mechanisms—specifically, the degeneration and aberrant hyper-regeneration and sprouting of DA axons. In contrast, the iPLA2-signaling pathway is thought to play a central role in regulating both monoamine axon degeneration and hyper-regeneration. Additionally, GSK3 signaling appears to contribute to axonal hyper-regeneration. As such, targeting iPLA2, GSK3, and their associated signaling pathways represents a promising therapeutic strategy for developing novel treatments for monoamine axon disorders and other neurodegenerative diseases, including Parkinson’s disease and Alzheimer’s disease.

## 7. Conclusions and Perspective

In conclusion, schizophrenia may be conceptualized as a disorder involving both the degeneration and hyper-regeneration of DA axons in the mPFC. The anti-inflammatory enzyme iPLA2 is proposed to play a critical role in driving hyper-regeneration, potentially contributing to the emergence of positive symptoms. To further evaluate this hypothesis, the following points warrant careful consideration in future research.

The symptoms of schizophrenia may be specifically associated with abnormal DA axons originating from the ventral tegmental area (VTA) and projecting to the mPFC. In contrast, major depression and bipolar disorder are considered disorders of broader monoaminergic systems, including NA, 5-HT, and DA axons [7,8]. NA axons from the locus coeruleus and 5-HT axons from the raphe nuclei are diffusely distributed throughout the brain, and their impairment likely contributes to the diverse symptoms observed in depressive disorders [8]. DA axons projecting from the VTA to the ventral striatum are thought to underlie symptoms, such as anhedonia and mania. Importantly, some patients exhibit co-occurring psychiatric symptoms, such as both positive and manic features. In such cases, it is hypothesized that DA axonal hyper-regeneration/sprouting occurs simultaneously in both the mPFC and ventral striatum. To better understand the symptom-region relationship, further studies using PET imaging of DA, NA, and 5-HT transporters are required to visualize monoaminergic axon terminals in patients with varying psychiatric symptoms. Additionally, advancements in imaging technology are necessary to achieve higher sensitivity and resolution, enabling more precise and comprehensive mapping of monoaminergic dysfunction.

Elevated serum levels of iPLA2 may serve as a useful biomarker for schizophrenia, particularly for predicting the onset of positive symptoms. Since overactivation of iPLA2 appears to drive axonal hyper-regeneration leading to these symptoms, monitoring serum iPLA2 levels could enable early intervention before symptom onset. Li, a well-known GSK3 inhibitor, has been reported to suppress hyperinnervation of sympathetic NA axons and may potentially reduce manic symptoms by inhibiting monoaminergic axon regeneration [7]. Accordingly, Li could be considered as a possible therapeutic approach for mitigating positive symptoms in schizophrenia. However, because high doses appear necessary to suppress aberrant axonal regeneration, careful monitoring would be essential to minimize the risk of toxic side effects. Interestingly, a recent study found that Li orotate—an organic Li salt—prevented age-related pro-inflammatory changes, synapse loss, and cognitive decline in mice without evident toxicity, even at much lower doses than the clinical standard, Li carbonate (an inorganic Li salt) [51]. If these findings can be validated in further studies, this newer form of Li salt may offer a safer strategy for limiting hyper-regeneration. Furthermore, pretreatment with iPLA2 inhibitors could also help prevent the onset of positive symptoms by restricting excessive DA axon sprouting. Despite these possibilities, the precise molecular mechanisms by which GSK3 and iPLA2 regulate axonal degeneration and regeneration remain unclear, warranting further investigation into their signaling pathways. Notably, during the transition from degeneration to hyper-regeneration, degenerative and hyper-regenerative axons may coexist. Such a condition could underlie the co-occurrence of negative and cognitive symptoms alongside positive symptoms, raising the challenging issue of how to treat both negative (degenerative) and positive (hyper-regenerative) symptoms simultaneously.

Given the possible association between negative symptoms and DA axon degeneration, promoting DA axon regeneration may represent a viable therapeutic approach. Antidepressants affecting the NA and 5-HT systems are known to facilitate axonal regeneration [11,52,53,54,55,56,57,58], potentially through iPLA2 activation. Accordingly, iPLA2 activators and their downstream metabolites, such as DHA and NPD1, may offer promising treatment strategies. Nonetheless, it is essential to assess baseline serum levels of iPLA2 and its metabolites prior to administering iPLA2-targeting agents.

The psychotomimetic effects of NMDA receptor antagonists (e.g., ketamine and phencyclidine), which induce both positive and negative/cognitive symptoms, suggest that NMDA receptor dysfunction contributes to schizophrenia. Under normal conditions, DA modulates NMDA activity in the mPFC: low DA concentrations enhance NMDA activity via D1R, while high concentrations inhibit it via D2R [45]. Excessive DA release in schizophrenia may lead to D2R overactivation, resulting in hyper-inhibition of NMDA function, which is thought to underlie positive symptoms [7]. However, the molecular pathways mediating D2R-mediated NMDA suppression remain poorly understood. Further research is needed to elucidate how NMDA signaling contributes to positive symptoms, particularly in the context of mPFC circuitry. Such insights may help uncover the neural mechanisms underlying psychotic symptoms like hallucinations and delusions.

Regarding negative symptoms, those observed in psychostimulant users are typically induced by NMDA receptor blockade, independent of DA transmission, and may involve D1R-expressing but not D2R-expressing neurons in the mPFC [7]. In contrast, the negative symptoms of schizophrenia are thought to result from reduced DA release due to axonal degeneration, affecting both D1R and D2R pathways (Figure 2). Thus, the nature of negative symptoms likely differs between psychostimulant-induced states and schizophrenia. Notably, negative symptoms involving D2R dysfunction may be unique to schizophrenia. Identifying such features could help distinguish schizophrenia-specific symptoms from those induced by drugs and clarify the role of D2R-mediated mechanisms in the disease.

Finally, the extent to which this new theory aligns with, expands upon, or challenges existing pathophysiological models remains an open question for future research.

## Figures and Tables

**Figure 1 cells-14-01348-f001:**
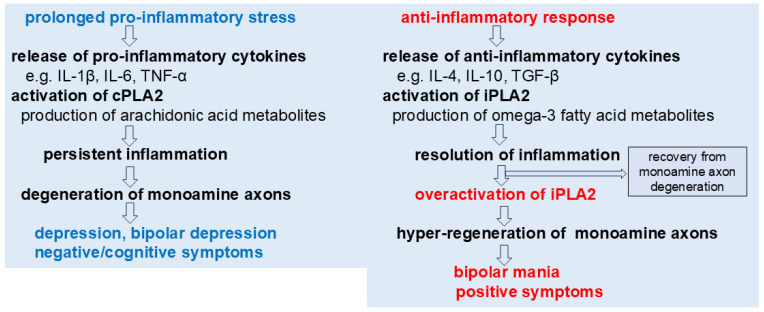
**A proposed role of phospholipase A2 in the symptoms of monoamine axon disorders.** cPLA2 may be involved in pro-inflammatory stress-induced degeneration of monoamine axons, potentially contributing to depression and negative/cognitive symptoms. In contrast, iPLA2 might participate in anti-inflammatory processes that support recovery from such degeneration. However, in certain conditions, excessive activation of iPLA2 could possibly drive hyper-regeneration of monoamine axons, which may underlie the emergence of bipolar mania and positive symptoms.

**Figure 2 cells-14-01348-f002:**
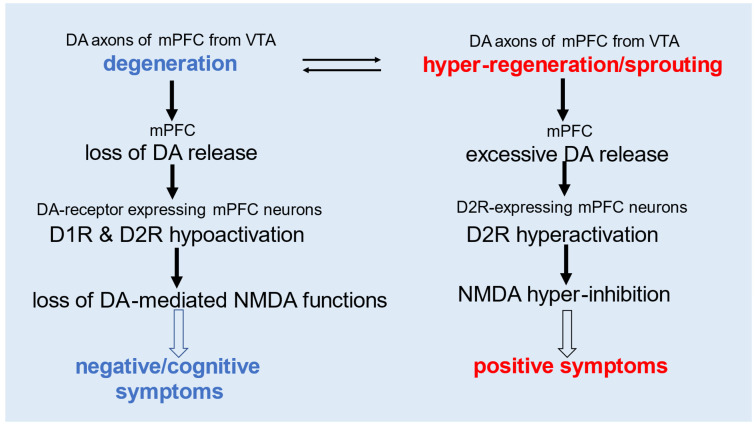
**Hypothesized mechanisms underlying negative/cognitive and positive symptoms.** Degeneration of DA axons in the mPFC may contribute to negative and cognitive symptoms by impairing both D1R- and D2R-mediated NMDA functions due to reduced DA release. In contrast, hyper-regeneration/sprouting of DA axons may lead to positive symptoms through D2R-mediated hyper-inhibition of NMDA activity, driven by excessive DA release. VTA: ventral tegmental area.

**Figure 3 cells-14-01348-f003:**
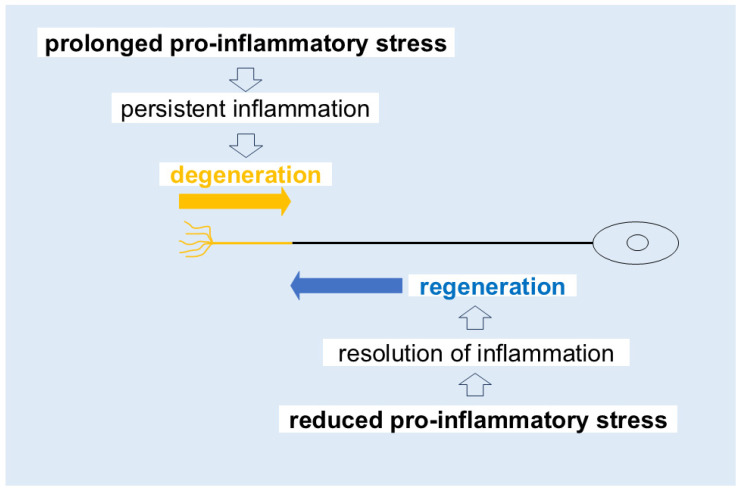
Pro- and anti-inflammatory competition in distal axons. Recovery from the symptoms of monoamine axon disorders is proposed to depend on the dynamic balance between pro- and anti-inflammatory responses, potentially reflecting the interplay between degenerative and regenerative mechanisms.

## Data Availability

No new data were created or analyzed in this study.

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
