# Peer review of "The Role of Calcium-Independent Phospholipase A2 in the Molecular Mechanisms of Schizophrenia"

_cells, 2025, doi:10.3390/cells14171348_

Round 1

Reviewer 1 Report

Comments and Suggestions for Authors

The author provides a review on monoamine axon degeneration and hyper-regeneration in schizophrenia, with special emphasis on Calcium-Independent Phospholipase A2. The topic - a deper dive into the pathophysiology of schizophrenia and a comparative view to other neuropsychiatric conditions is of general interest for a broader audience. 

At the same time, the manuscript appears selective in the data presented, leaving the impression that this could be the only pathomechanism. To the knowledge of this reviewer, there is no consensus that monoamine axon degeneration or hyperegeneration is seen as the only or deciding pathomechanism in schizophrenia. 
The whole manuscript would benefit from a better link to the existing theories, and a more cautious presentation. 

Specifically the following points are most critical, and may help the author to improve the manuscript: 

  1. The introduction starts with "Accumulating evidence suggests that, similar to early-stage Parkinson’s disease, depression is a neurodegenerative disorder characterized by the degeneration of monoamine axons without accompanying cell death". This and the following statements are referenced with publications by the author, both of them reviews. In the opinion of this reviewer, the cited references would mandate a more careful wording. 
  2. There is convincing evidence that the disorder is characterized by lower levels of glutamate, GABA and dopamine in the prefrontal cortex and other cortical regions, as well as alterations in glutamate and dopamine in the basal ganglia. Additionally there are quite some data suggesting an imbalance in glutamate and GABA in the pathophysiology. On top there is a wealth of data on structural and functional findings (connectome, including a potentially leading role key role for fronto-thalamo-striatal–midbrain circuits). The review falls short in linking the research question to the overall field. 
  3. Figure 1: For this reviewer, it is unclear on which studies and original publications the figure is based on. Is this a hypothesis, or is this based on existing data? If based on data, on which studies? 
    The same holds true for figure 2. 
  4. The author formulates the hypothesis (?), that positive symptoms are caused by hyper-regeneration/ sprouting of monoamine axons, while negative/cognitive symptoms are caused by degeneration of monoamine axons. This would be linked to different inflammatory mechanisms. It remains open (at least in the text) how these processes occur at the same time in the same brain.  
  5. Line 201 - 211, plus rest of section 6: While the statements are quite bold, there are no references supporting them. It reamins unclear, whether the claimed  "increased recognition of  schizophrenia, major depression, and bipolar disorder as neurodegenerative conditions associated with the degeneration of distal monoamine  neuron axons" is the opinion of the author, or really a widely accepted consensus in the scientific and medical community. 
  6. The conclusion appears too bold with respect to the complex pathophysiology of schizophrenia. 

This reviewer suggests to use these points as guiding principles when working on the next version. There are quite a few points in the same direction in the paper, which would be too many to list them individually here. 

Reviewer 2 Report

Comments and Suggestions for Authors

This is well developed paper I however have some comments.

  • I the introduction include a more comprehensive background on the role of iPLA2 in neurodegenerative and psychiatric disorders, including its biochemical function its role in releasing arachidonic acid from phospholipids and its relevance to neuronal integrity.
  • Better define terms like "monoamine axons" and specify the neurotransmitter systems dopamine, serotonin, norepinephrine implicated.
  • Include a brief overview of existing literature on iPLA2 in schizophrenia to establish the novelty of the perspective.
  • The authors theorizes that overactivation of iPLA2 leads to hyper-regeneration of monoamine axons, contributing to positive symptoms, while proinflammatory iPLA2 activation drives degeneration linked to negative symptoms. This is an interesting hypothesis, but it lacks detailed mechanistic explanations. To enhance this, I suggest to elaborate on how iPLA2 overactivation leads to hyper-regeneration through downstream effects on lipid signaling, neurotrophic factors like NGF, or cytoskeletal remodeling.
  • Specify the molecular pathways linking proinflammatory iPLA2 activation to axon degeneration.
  • The conclusion briefly mentions targeting iPLA2 and GSK3 as a promising therapeutic strategy but does not elaborate on practical implications or challenges. For pharama discuss specific iPLA2 inhibitors (bromoenol lactone) or GSK3 inhibitors (lithium, as referenced in Shah et al.) and their potential efficacy or limitations in treating schizophrenia.
  • Address potential side effects or off-target effects of modulating iPLA2, given its broad role in lipid metabolism and inflammation.
  • Propose future research directions, such as clinical trials to test iPLA2 inhibitors in schizophrenia or studies to validate the axonal regeneration hypothesis in patient-derived models.
  • The acknowledgment of ChatGPT for improving the English is awkward.

Reviewer 3 Report

Comments and Suggestions for Authors

This perspective article proposes that schizophrenia, depression, and bipolar disorder should be conceptualized as monoaminergic axon disorders, involving both degeneration and aberrant regeneration. The author highlights the role of calcium-independent phospholipase A2 (iPLAâ‚‚) as a key modulator of anti-inflammatory processes and axonal sprouting, potentially contributing to positive symptoms of schizophrenia. The review also examines dopamine-NMDA receptor interactions in the medial prefrontal cortex (mPFC) and proposes iPLAâ‚‚ as a novel therapeutic target. The model integrates neuropathological, biochemical, and developmental data to build a unified framework for schizophrenia pathophysiology. While innovative, it remains largely theoretical and would benefit from clearer boundaries between speculation and evidence.

Abstract

The abstract includes several strong claims (e.g., "schizophrenia as a monoaminergic axon disorder") that may overreach current evidence. A qualifying statement indicating the speculative nature of this framework would improve  balance.

Introduction

The introduction effectively sets the stage by drawing parallels with Parkinson's disease and highlighting monoaminergic axon degeneration in schizophrenia. The comparison is intriguing but should be better supported by direct clinical evidence, particularly for bipolar disorder. The author might consider softening statements such as "schizophrenia may be conceptualized as..." to "can be hypothetically conceptualized as..." to acknowledge the theoretical nature.

Section 2: Cortical atrophy and negative/cognitive symptoms

This section appropriately reviews postmortem findings supporting dopaminergic axon degeneration in the mPFC. However, the interpretation that degeneration directly maps onto negative/cognitive symptoms should be more cautious, as causality remains difficult to establish. The cited study (Akil et al.) is relevant but limited by small sample size and variability across cases. The extrapolation to bipolar depression is under-referenced and could benefit from additional literature.

Section 3: iPLAâ‚‚ and positive symptoms

This is the central section of the article and provides an interesting hypothesis on iPLAâ‚‚'s dual role in axonal regeneration and symptom emergence. The leap from biochemical findings to behavioral symptoms (e.g., paranoia and axon branching) lacks mechanistic clarity and should be presented with more nuance. The role of glutamatergic and GABAergic axons is briefly mentioned but underdeveloped; if included, it warrants further elaboration or referencing.

Section 4: Peripheral hyperinnervation

The extension of findings to peripheral tissues is conceptually coherent and biologically plausible, especially regarding NGF and GSK3. However, the clinical relevance of these peripheral phenomena to schizophrenia is speculative. This must be clearly evidenced and sentences must be more cautious.

Section 5: mPFC and DA/NMDA interactions

This section builds a compelling neurodevelopmental narrative by linking neonatal NRG1 exposure to mPFC DA hyperinnervation. The distinction between D1/D2 receptor-mediated NMDA modulation is well presented. Nonetheless, the model combining axonal plasticity with NMDA hypofunction remains speculative, and the article would benefit from discussing potential confounders (e.g., local interneuron dysfunction, oxidative stress, etc.).

Section 6: Diagnostic and therapeutic implications

The discussion on PET imaging to assess monoaminergic axons is valuable but should acknowledge its current limitations in clinical practice. The therapeutic implications involving iPLAâ‚‚ and GSK3 pathways are innovative, yet concrete evidence from clinical trials or drug development is lacking. Acknowledging this gap is mandatory.

Conclusion

The conclusion is concise but should reiterate that the model is speculative and based on converging indirect evidence. The acknowledgment of ChatGPT in editing is acceptable but might be more appropriately phrased as "for assistance in English editing."

The limitations of the proposed model are insufficiently addressed. Key concerns include:

The speculative nature of linking iPLAâ‚‚ activity to positive symptoms without direct causal evidence.

Lack of longitudinal or human experimental data to support the proposed regenerative mechanism.

Overgeneralization from peripheral to central nervous system mechanisms.

Future directions might include the use of induced pluripotent stem cell (iPSC) models or in vivo imaging of axonal regeneration in patients

Round 2

Reviewer 1 Report

Comments and Suggestions for Authors

This reviewer acknowledges the relevant improvements made by the author. At the same time there are still points that may improve the paper: 

  1. In answer 4, the author suggests a potentially causal link between a pro-inflammatory respnse leading to axon degeneration and negative & cognitive symptoms and a subsequent antiinflammatory response with hyperregeneration. If these 2 processes are thought to be linked, this should be reflected in the figure. 
    This reviewer further suggests to substantiate this further, and critically reflect on open questions. 
    In  principle, the sequence of events with cognitive and negative symptoms starting prior to positive symptoms may support the proposed pathophysiological mechanism. 
    However, negative and cognitive symptoms may precede the onset of positive symptoms by years. And there is no indication that negative or cognitive symptoms improve after the onset of positive symptoms - most often they remain unchanged or worsen. 
  2. The author refers to NMDA agonists as treatment for all symptom domains of schizophrenia (e.g. line 206, 249) without references. Today, drugs acting on the NMDA receptor are not used as firstline treatment in schizophrenia, and more commonly as second or third line in MDD. A more careful description would be benefitial. 
  3. The conclusions and perspectives have greatly improved, however they offer a new hypothesis now (use of Li in different salts as potential therapeutic), which was not discussed in the review (this would rather fit e.g. in section 5). In the opinion of this reviewer, it would be benefitial to conclude with a more concise view on what are the strongest points supporting the hypotheses presented in this article, and what are the main open research questions. One might finish with an outlook on therapeutic opportunities.  
    This also includes to briefly discuss potential linking points to other pathophysiological theories, as already mentioned in point 2 of the first review round. 
  4. Regarding comment 5, the new sentence is only slightly better. “Schizophrenia, major depression, and bipolar disorder are thought to be 
    neurodegenerative conditions associated with the degeneration of 
    monoaminergic axons.” This reviewer suggests to use conditional forms if there is no reference for a consensus in the scientific community (e.g. may be thought of, ...). 

Reviewer 2 Report

Comments and Suggestions for Authors

None

Author Response

Thank you.

Reviewer 3 Report

Comments and Suggestions for Authors

Thank you for your work

Author Response

Thank you

Round 3

Reviewer 1 Report

Comments and Suggestions for Authors

This reviewer acknowledges the detailed explanations by the author, making it more clear on the viewpoints. 
At the same point, the questions are not for the reviewer, but for any potential reader. This reviewer encourages the author to be transparent about open questions.  

1) Maybe the author may add one sentence like "While positive, negative and cognitive symptoms occur simultanously, the temporal sequence and frequency of degenerative and hyper-regenerative axons over the course of the disease remains an open question."  (just as an example)

3) Again, a small point in the discussion would be great stating that how this new theory aligns/expands or replaces current pathophysiological theories reamins an open research question.  
